# Erasable labeling of neuronal activity using a reversible calcium marker

**Fern Sha, Ahmed S Abdelfattah, Ronak Patel, Eric R Schreiter***

Howard Hughes Medical Institute, Janelia Research Campus, Ashburn, United States

**Abstract** Understanding how the brain encodes and processes information requires the recording of neural activity that underlies different behaviors. Recent efforts in fluorescent protein engineering have succeeded in developing powerful tools for visualizing neural activity, in general by coupling neural activity to different properties of a fluorescent protein scaffold. Here, we take advantage of a previously unexploited class of reversibly switchable fluorescent proteins to engineer a new type of calcium sensor. We introduce rsCaMPARI, a genetically encoded calcium marker engineered from a reversibly switchable fluorescent protein that enables spatiotemporally precise marking, erasing, and remarking of active neuron populations under brief, user-defined time windows of light exposure. rsCaMPARI photoswitching kinetics are modulated by calcium concentration when illuminating with blue light, and the fluorescence can be reset with violet light. We demonstrate the utility of rsCaMPARI for marking and remarking active neuron populations in freely swimming zebrafish.

**\*For correspondence:**
schreitere@janelia.hhmi.org

## Introduction

Important behaviors such as response to stimuli and memory retrieval are governed by patterns of coordinated neuron activity in the brain. Identifying which neuron ensembles are involved and how they compare across different behavior states is critical for our understanding of brain circuitry and function. Traditional methods for identifying active neuron ensembles have relied on exploiting the cellular machinery that control expression of immediate early genes (IEGs) such as *Egr1/Zif268*, *c-Fos*, or *Arc* (*Morgan et al., 1987*; *Cole et al., 1989*; *Sheng and Greenberg, 1990*; *Lyford et al., 1995*; *Kawashima et al., 2014*). However, the temporal resolution of these methods is poor, typically on the scale of tens of minutes to hours, and they do not correlate well with electrical activity (*Sallaz and Jourdan, 1993*; *Sheng et al., 1993*). More accurate and temporally precise methods are desired for identifying and comparing active neuron ensembles.

Calcium is an important second messenger in cell signaling pathways and neuron activity (*Grienberger and Konnerth, 2012*), which follows electrical activity more quantitatively (*Helmchen et al., 1997*; *Bajar et al., 2016*). Over the past several decades, a wide range of tools have been developed to monitor and record calcium activity in cells, tissues, and live animals. For example, GCaMPs (*Tian et al., 2009*; *Akerboom et al., 2012*; *Chen et al., 2013*) are single-wavelength calcium indicators that have enabled the visualization of calcium transients with fast kinetics, but their requirement for constant monitoring limit the size of the field of view, and recording neural activity in freely moving specimens is challenging. CaMPARIs (*Fosque et al., 2015*; *Moeyaert et al., 2018*) are photoconvertible calcium integrators that have enabled optical selection and post-hoc analysis of the evoked calcium activity across large tissue volumes during a specific epoch of interest. Alternative approaches have coupled calcium activity with light activation in order to drive the activation of transcription factors (*Lee et al., 2017*; *Wang et al., 2017*). These calcium sensors are powerful tools for identifying and comparing active neuron ensembles and there is continued demand for tools with improved or novel properties.

One major limitation of calcium sensors that rely on photoconversion or light activation is their irreversibility, which is useful when a permanent signal is desired; however, this irreversibility also hinders reuse of the tool, especially when multiple snapshots of activity are desired from the same sample preparation. Multiple snapshots are desirable when comparing active neuron ensembles in non-stereotyped organisms where there is variability in neural circuitry across different individuals, or where there is variability from trial to trial in the same individual. Thus, one desirable property for a calcium sensor would be the ability to mark, erase, and then re-mark calcium activity during specified time periods over large scales. Such an erasable calcium marker would enable multiple snapshots of neuronal activity to be captured quickly within the same sample preparation.

To develop an erasable calcium marker, we used protein engineering to combine calcium sensing with the properties of reversibly switchable fluorescent proteins. These proteins can be toggled between a fluorescent state (on) and a non-fluorescent state (off) depending on the wavelength of light used for illumination (*Zhou and Lin, 2013*). For example, Dronpa (*Ando et al., 2004*), rsEGFP (*Grotjohann et al., 2011*), and mGeos (*Chang et al., 2012*) are bright green fluorescent proteins that can be off-switched with blue light illumination. Subsequent illumination with violet light reverts them back to the on-state. The structural mechanism underlying this reversible switching behavior is a cis/trans isomerization of the chromophore, which is coupled to the chromophore's protonation state (*Andresen et al., 2005*; *Andresen et al., 2007*).

We hypothesized that reversible photoswitching could provide a mechanism for marking, erasing, and remarking calcium activity using light. Specifically, we aimed to engineer an erasable calcium marker such that the photoswitching kinetics are $Ca^{2+}$-dependent and that the final magnitude of fluorescence change provides a stable but erasable readout for the relative amount of calcium activity during a user-defined time window of light illumination; reverse photoswitching could then quickly erase the signal and reset the tool for immediate reuse. Here, we report development of a new type of erasable calcium marker called rsCaMPARI (reversibly switchable CaMPARI)(*Figure 1a*). Using directed evolution methods, we engineered the off-switching kinetics of rsCaMPARI under blue light illumination to be $Ca^{2+}$-dependent, which allows reversible calcium activity marking with cellular resolution. The fluorescence can be easily recovered with violet light for immediate reuse and enables erasable and repeatable calcium activity marking over practical timescales of a few seconds. We demonstrate the utility of rsCaMPARI for marking, erasing, and re-marking cells with elevated calcium in primary neuron cultures and freely swimming zebrafish.

## Results

### Engineering and in vitro characterization of rsCaMPARI

To engineer an erasable calcium marker, we took inspiration from CaMPARI and CaMPARI2 since they were previously shown to function well in several in vivo preparations (*Fosque et al., 2015*; *Moeyaert et al., 2018*). We introduced CaMPARI2 mutations onto a mEos3.1 (*Zhang et al., 2012*) scaffold and converted the protein from a green-to-red photoconvertible FP to a reversibly switchable FP that cycles between bright green and dim states via an H62L substitution within the chromophore, as was previously demonstrated by the production of the mGeos FPs from EosFP (*Chang et al., 2012*). After photoswitching to the off-state, the H62L mutant exhibited slow spontaneous recovery to the on-state with a half-time of several hours (*Chang et al., 2012*). We reasoned that the stability of the switched fluorescent signal would be beneficial for stable and precise readout of the final fluorescence signal following photoswitching.

We next engineered libraries of insertions of a calcium-binding domain (calmodulin) and calmodulin-binding peptide into the H62L-substituted FP (*Figure 1—figure supplement 1*). Guided by crystal structures of CaMPARI (*Fosque et al., 2015*) and the cis/trans isomers of IrisFP (*Adam et al., 2008*), we targeted the insertion of calcium-binding domains to the middle of β-strands 8 and 9 to efficiently propagate $Ca^{2+}$-induced conformational changes to amino acids around Y63 of the chromophore, which undergoes cis/trans isomerization during reversible photoswitching. The libraries included both possible orientations of the calcium-binding domains and had diversity in the length and composition of linkers connecting the calcium-binding and FP domains. A red fluorescent protein, mCherry (*Shaner et al., 2004*), was fused in frame at the C-terminus to normalize for expression and photoswitching. Screening in *E. coli* lysate (*Figure 1—figure supplement 2*), we aimed to

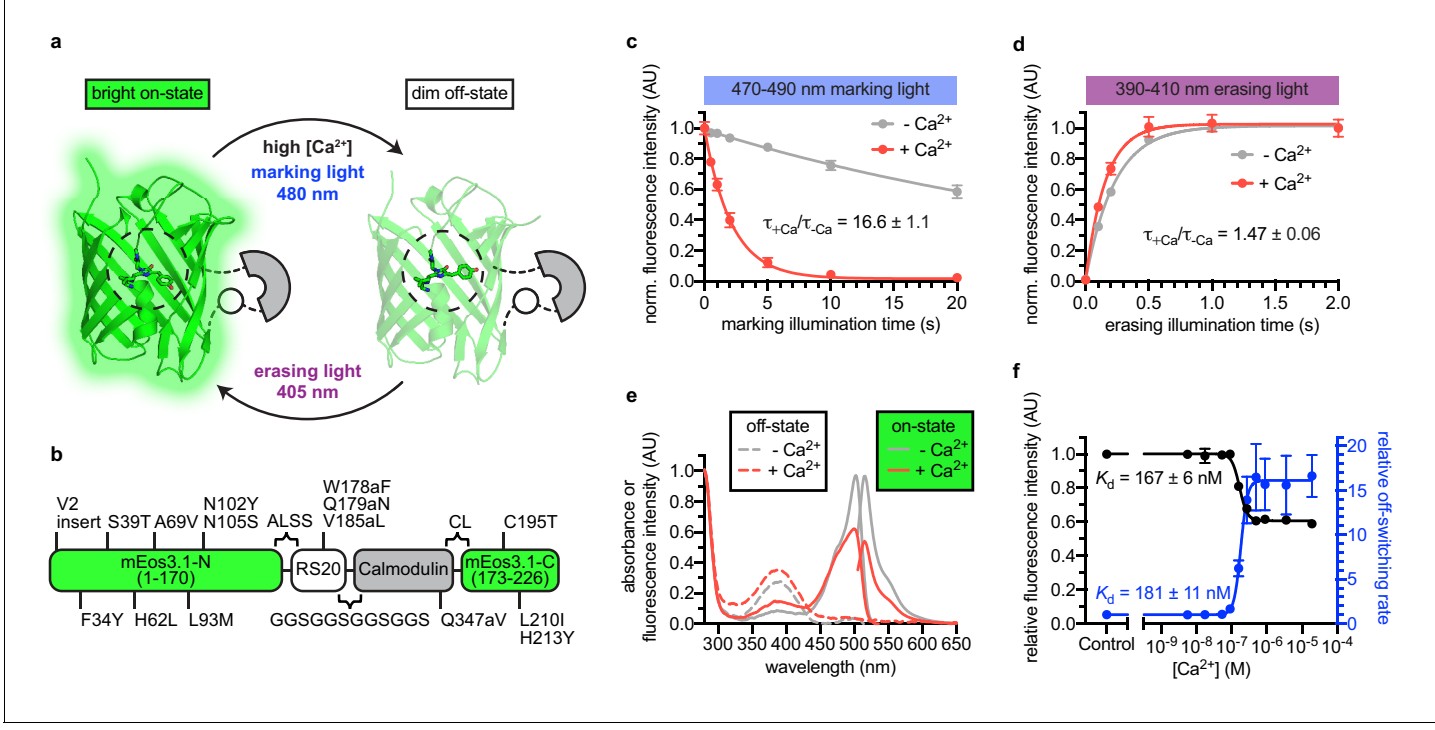

**Figure 1.** Engineering and in vitro characterization of rsCaMPARI. (**a**) Schematic of rsCaMPARI function. (**b**) Primary structure of rsCaMPARI relative to mEos3.1. (**c**) Off-switching time-course of rsCaMPARI under marking light illumination (200 mW/cm$^2$) in the presence or absence of calcium. Lines are single-exponential fits to data. Error bars are standard deviation, n = 4 replicate measurements. (**d**) On-switching time-course of rsCaMPARI under erasing light illumination (200 mW/cm$^2$) in the presence or absence of calcium. Lines are single-exponential fits to data. Error bars are standard deviation, n = 4 replicate measurements. (**e**) Spectral properties of rsCaMPARI. Absorbance or fluorescence emission spectra of rsCaMPARI in the fluorescent on-state in the presence or absence of calcium shown with solid traces. Absorbance spectra of rsCaMPARI in the non-fluorescent off-state in the presence or absence of calcium shown with dashed traces. (**f**) Relative fluorescence intensity and relative off-switching rate of rsCaMPARI as a function of free [Ca$^{2+}$]. Lines are sigmoidal fits to data. Error bars are standard deviation, n = 4 replicate measurements.

The online version of this article includes the following source data and figure supplement(s) for figure 1:

**Source data 1.** Off-switching time-course of rsCaMPARI under marking light illumination in the presence or absence of calcium.

**Source data 2.** On-switching time-course of rsCaMPARI under erasing light illumination in the presence or absence of calcium.

**Source data 3.** Relative fluorescence intensity and relative off-switching rate of rsCaMPARI as a function of free [Ca$^{2+}$].

**Figure supplement 1.** Design and construction of libraries to engineer an erasable calcium activity marker.

**Figure supplement 2.** Outline of library screening to engineer rsCaMPARI.

**Figure supplement 3.** In vitro characterization of variants selected from library screening.

**Figure supplement 4.** DNA and amino acid sequence of rsCaMPARI with sequence features annotated.

**Figure supplement 5.** Amino acid sequence comparison of rsCaMPARI with mEos3.1.

**Figure supplement 6.** rsCaMPARI secondary library screen showing relationship between photoswitching contrast and indicator behavior.

**Figure supplement 7.** Two-photon action cross-section of rsCaMPARI in the presence and absence of Ca$^{2+}$.

**Figure supplement 8.** Optimization of blue light wavelengths for off-switching of rsCaMPARI.

**Figure supplement 8—source data 1.** Off-switching time-course of rsCaMPARI in the presence or absence of calcium under various wavelengths of light illumination.

**Figure supplement 8—source data 2.** Off-switching rate contrast and maximum fluorescence change (ΔF) as a function of wavelength.

**Figure supplement 8—source data 3.** On-switching time-course of rsCaMPARI from dim off-state under illumination with 460 nm light.

**Figure supplement 9.** The relationship between rsCaMPARI off-switching rate and light power intensity.

**Figure supplement 9—source data 1.** The relationship between rsCaMPARI off-switching rate and light power intensity.

optimize four parameters: (1) difference in green fluorescence following 490 nm light illumination +/- Ca$^{2+}$, (2) recovery of fluorescence intensity following 400 nm light illumination, (3) minimum fluorescence change due to calcium binding in the absence of light illumination ('indicator behavior'), and (4) green brightness. Ninety-six clones were selected and sequenced, resulting in 19 unique clones that exhibited a variety of photoswitching kinetics (*Figure 1—figure supplement 3*).

One variant with an insertion of calcium-binding domains in β-strand nine was selected for further characterization due to its > 10 fold photoswitching rate contrast, relatively small indicator behavior, and high brightness. We named this variant rsCaMPARI (reversibly switchable CaMPARI) (*Figure 1*, *Figure 1—figure supplement 4*, and *Figure 1—figure supplement 5*). Additional site-saturation mutagenesis to improve rsCaMPARI contrast did not identify improved variants without compromising other desired traits. In particular, we observed that any improved fluorescence contrast appeared to be tightly coupled to increased indicator behavior (*Figure 1—figure supplement 6*).

rsCaMPARI is a bright green and calcium-dependent photoswitchable FP (*Figure 1*, *Table 1*, and *Figure 1—figure supplement 7*). We observed a strong dependence of the extent of photoswitching and the rate contrast on the wavelength of light applied (*Figure 1—figure supplement 8a–d*). During illumination with blue 'marking light' (470–490 nm, 200 mW/cm$^2$, *Figure 1—figure supplement 8e*), rsCaMPARI exhibited ~17 fold faster off-switching kinetics in high calcium compared to low calcium (*Figure 1c*) and the off-switching rate was linear with respect to the light power (*Figure 1—figure supplement 9*). When exposed to violet 'erasing light' (390–410 nm, 200 mW/cm$^2$), the recovery of rsCaMPARI fluorescence back to the bright on-state was very efficient and had similar on-switching kinetics in either high- or low-calcium conditions (*Figure 1d*). Measurement of green fluorescence intensity as a function of free calcium concentration showed that the calcium-free state was ~2-fold brighter than the calcium-bound state (*Figure 1e,f*), and indicated an apparent dissociation constant of 167 ± 6 nM (*Figure 1f*). Measurement of the off-switching rate vs. free calcium concentration gave a similar apparent dissociation constant of 181 ± 11 nM (*Figure 1f*). The calcium affinity of rsCaMPARI is therefore similar to previously described CaMPARIs (*Fosque et al., 2015*; *Moeyaert et al., 2018*) and GCaMPs (*Tian et al., 2009*; *Akerboom et al., 2012*; *Chen et al., 2013*), suggesting that the dynamic range of rsCaMPARI calcium sensitivity falls within the physiological range of neuronal calcium transients.

## Characterization of rsCaMPARI in dissociated neurons

We expressed rsCaMPARI in dissociated primary rat hippocampal neuron . The mCherry tag was replaced with mRuby3 (*Bajar et al., 2016*). mCherry is derived from DsRed, which is known to accumulate and form bright puncta in cells (*Katayama et al., 2008*). We used field electrode stimulation to induce action potential firing at 80 Hz during continuous illumination with marking light (200 mW/cm$^2$) on an epifluorescence microscope (*Figure 2a–e* and *Figure 2—figure supplement 1*). Neurons illuminated with 10 s of marking light accompanied by field stimulation exhibited strong dimming of the green fluorescence signal (*Figure 2a*, cycle 1), which could later be recovered by a 3-s pulse of erasing light. When the same neuron was subsequently illuminated with marking light but without field stimulation, only modest dimming of the green fluorescence was observed (*Figure 2a*, cycle 2). The same neuron could be repeatedly marked and erased over several cycles (*Figure 2a*, cycles 3–6). To assess rsCaMPARI sensitivity, we quantified green fluorescence changes during a 2 s pulse of marking light over a range of stimulation frequencies from 5 to 80 Hz (*Figure 2b*). While the non-stimulated neurons exhibited minimal photoswitching, we observed significantly more photoswitching in stimulated neurons and higher frequencies resulted in more photoswitching. The broad variability we observed in the marked signal is presumably due to the natural variability among neurons and this variability was previously observed for GCaMP in similar neuron preparations (*Wardill et al., 2013*). Monitoring green fluorescence of the neurons over time showed a shift to more rapid off-

**Table 1.** Photophysical properties of rsCaMPARI.

| | Ca$^{2+}$ | λ$_{abs}$ (nm) | λ$_{ex}$ (nm) | ε (M$^{-1}$cm$^{-1}$) | Φ$_F$ | Brightness* | ↓ rate$^†$ (s$^{-1}$) | ↑ rate$^‡$ (s$^{-1}$) | $K_d$ (nM) | Hill coefficient |
|---|---|---|---|---|---|---|---|---|---|---|
| rsCaMPARI (on) | - | 502 | 515 | 60304 | 0.54 | 32.6 | 0.028 ± 0.002 | - | 167 ± 6 | 3.96 |
| | + | 500 | 515 | 38234 | 0.46 | 17.6 | 0.467 ± 0.066 | - | | |
| rsCaMPARI (off) | - | 392 | - | - | - | - | - | 4.37 ± 0.24 | - | - |
| | + | 389 | - | - | - | - | - | 6.40 ± 0.28 | | |

*Brightness = ε x Φ$_F$ / 1000.
$^†$Measured during irradiation with 200 mW/cm$^2$ of 484 nm light.
$^‡$Measured during irradiation with 200 mW/cm$^2$ of 405 nm light.

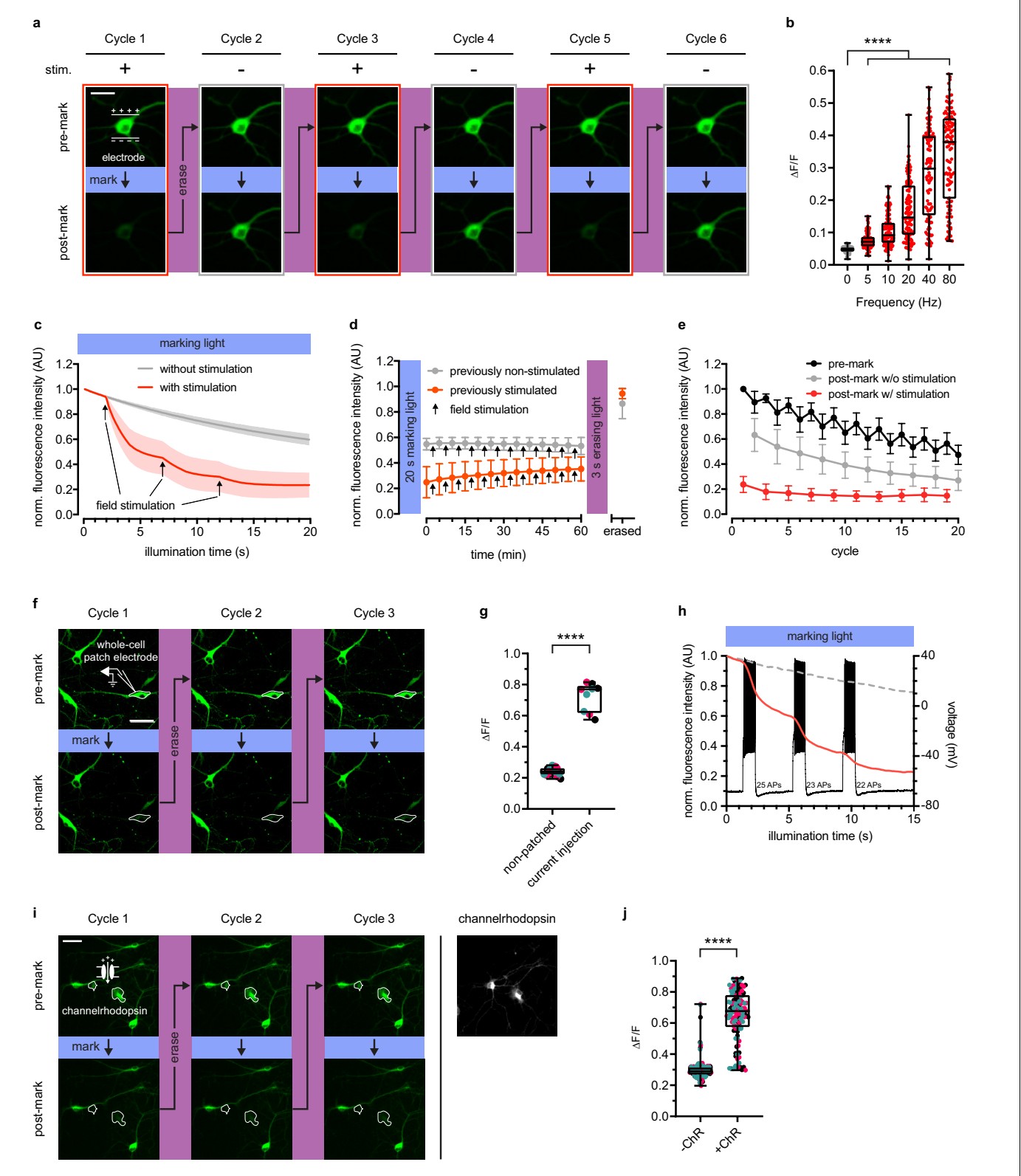

**Figure 2.** rsCaMPARI reversibly and selectively marks activity in primary neurons. (**a**) Fluorescence images of a representative primary rat hippocampal neuron undergoing multiple cycles of exposure to a 10 s window of marking light (224 mW/cm$^2$) ± field stimulation (2 × 160 stimulations at 80 Hz). Each cycle was reset with a 3-s pulse of erasing light (224 mW/cm$^2$). Scale bar is 30 μm. (**b**) Quantification of ΔF/F of individual neurons (n = 119 neurons from four independent wells) ± field stimulations from 5 to 80 Hz during a 2-s pulse of marking light. Boxplot whiskers extend from minimum to maximum

*Figure 2 continued on next page*

*Figure 2 continued*

values and box extends from 25th to 75th percentile. Middle line in box is the median. ****p<0.0001, two-tailed Student's *t*-test. (c) Fluorescence time-course traces of neurons undergoing one cycle of illumination ± stimulation (3 × 160 stims at 80 Hz). Arrows on time-course trace denote start of each stimulation. Error bars are standard deviation, n = 15 neurons from two independent wells. (d) Time-course of rsCaMPARI spontaneous recovery in the dark at 37°C following marking light illumination using previously non-stimulated or previously stimulated neurons prepared under similar conditions as (c). Arrows denote a bout of field stimulation between each imaging timepoint (160 stims at 80 Hz). After 1 hr, the neurons were reset with a 3-s pulse of erasing light. Error bars are standard deviation, n = 66 previously non-stimulated neurons and n = 59 previously stimulated neurons from two independent wells. (e) Photofatigue of rsCaMPARI over successive cycles of marking light illumination with or without field stimulation. Each cycle is followed by erasing light to reset the sensor. Error bars are standard deviation, n = 7 neurons from three independent wells. (f) Fluorescence images of rsCaMPARI before and after 15 s of marking light illumination (150 mW/cm$^2$). A single cell, denoted by pipette drawing, was patched and stimulated to fire action potentials by injecting current during marking light illumination. Scale bar is 50 µm. (g) Quantification of ΔF/F of individual neurons across three marking cycles for patched (n = 3 neurons from three independent wells) and non-patched cells (n = 8 neurons from three independent wells). Cyan, red, and black data points are measurements from the first, second, and third cycles, respectively. ****p<0.0001, two-tailed Student's *t*-test. (h) Single-trial recording of action potentials from current injection during the first cycle in patched neuron shown in (f) using fluorescence imaging (red trace) or electrophysiology to measure membrane potential (black trace). Average fluorescence traces of non-patched neurons are shown as dashed grey trace. (i) Fluorescence images of rsCaMPARI before and after 10 s of marking light illumination (285 mW/cm$^2$) (left panels). Two neurons denoted by a white outline are positive for channelrhodopsin (ChR) expression. Right-most panel is a fluorescence image of channelrhodopsin expression. Scale bar is 100 µm. (j) Quantification of ΔF/F of individual neurons across three marking cycles for +ChR (n = 42 neurons from 17 independent wells) and -ChR (n = 79 neurons from 17 independent wells) cells. Cyan, red, and black data points are measurements from the first, second, and third cycles, respectively. ****p<0.0001, Wilcoxon rank-sum test.

The online version of this article includes the following source data and figure supplement(s) for figure 2:

**Source data 1.** ΔF/F values of individual neurons ± field stimulations from 5 to 80 Hz during a 2-s pulse of marking light.

**Source data 2.** Fluorescence time-course of neurons undergoing one cycle of illumination ± stimulation (3 × 160 stims at 80 Hz).

**Source data 3.** Time-course of rsCaMPARI spontaneous recovery in the dark at 37°C following marking light illumination.

**Source data 4.** Photofatigue of rsCaMPARI over successive cycles of marking light illumination with or without field stimulation.

**Source data 5.** Quantification of ΔF/F of individual neurons across three marking cycles for patched and non-patched cells.

**Source data 6.** Fluorescence time-course of rsCaMPARI in patched and non-patched cells during marking light illumination.

**Source data 7.** Quantification of ΔF/F of individual neurons across three marking cycles for +ChR and -ChR cells.

**Figure supplement 1.** rsCaMPARI marks neurons stimulated by a field electrode (data from *Figure 2a and c–e* normalized using mRuby3).

**Figure supplement 1—source data 1.** Fluorescence time-course of neurons undergoing one cycle of illumination ± stimulation (3 × 160 stims at 80 Hz).

**Figure supplement 1—source data 2.** Time-course of rsCaMPARI-mRuby3 spontaneous recovery in the dark at 37°C following marking light illumination.

**Figure supplement 1—source data 3.** Photofatigue of rsCaMPARI-mRuby3 over successive cycles of marking light illumination with or without field stimulation.

**Figure supplement 2.** Spontaneous recovery of rsCaMPARI.

**Figure supplement 2—source data 1.** Time-course of rsCaMPARI-mRuby3 spontaneous recovery in the dark at 37°C following marking light illumination.

**Figure supplement 3.** rsCaMPARI selectively marks neurons stimulated by current injection through a patch pipette.

**Figure supplement 4.** rsCaMPARI selectively marks neurons activated by a channelrhodopsin (data from *Figure 2i–j* normalized using mRuby3).

**Figure supplement 4—source data 1.** Logarithm values of the red-to-green fluorescence ratios of -ChR and +ChR neurons post-marking light illumination.

**Figure supplement 4—source data 2.** The relationship between log(red/green) values and ΔF/F.

switching that correlated with the onset of field stimulation (*Figure 2c*). In response to three trains of 160 action potentials during marking light, we observed the largest fluorescence change during the first train (~40–60% green fluorescence decrease compared to ~10% in non-stimulated neurons). Smaller fluorescence changes were observed during the second and third field pulse trains as the sensor approached near complete off-switching.

In order for rsCaMPARI activity marking to be compatible with post-hoc analysis, it is important that the post-marking fluorescence is stable. We tracked the fluorescence of neurons that had previously been illuminated with marking light and found that rsCaMPARI green fluorescence is stable in the dark for more than 1 hr (*Figure 2d*, *Figure 2—figure supplement 1c*, and *Figure 2—figure supplement 2*), even during ongoing calcium activity. This provides sufficient time for readout of the recorded signal, for example with high-resolution microscopy techniques over large volumes of tissue, and demonstrates that rsCaMPARI is sufficiently stable for post-hoc analysis.

Next, we characterized photofatigue of rsCaMPARI in neurons from multiple rounds of marking and erasing to understand how many times activity patterns could be marked. Fluorescence was monitored before and after marking light illumination, then recovered with erasing light over successive cycles (*Figure 2e*). The neurons were stimulated during odd-numbered cycles, but not during even-numbered cycles. We observed that rsCaMPARI lost ~50% of its initial fluorescence after 20 cycles. The fluorescence contrast between 'stimulated' and 'not stimulated' was reduced to ~50% of the initial difference after 10 cycles, but still allowed discrimination between marked and unmarked cells. rsCaMPARI may therefore be erased and reused for at least 10 cycles of marking activity.

To demonstrate that rsCaMPARI can mark an active subset of neurons within a population, we first performed whole-cell patch clamp electrophysiology on a single rsCaMPARI-expressing neuron, delivering current injections to produce controlled action potential firing during marking light (*Figure 2f–h* and *Figure 2—figure supplement 3*). Other neurons in the same field of view were not active due to application of drugs to block synaptic release. The patched neuron became much dimmer during the marking light illumination (*Figure 2f*, bottom panels), providing threefold contrast when compared to surrounding neurons that were not patched (*Figure 2g*), and more rapid off-switching correlated well with action potential firing (*Figure 2h*). The green rsCaMPARI fluorescence could be quickly recovered by subsequent illumination with erasing light and selective marking of the patched neuron could be repeated several times (*Figure 2f*, cycles 2 and 3). We also drove activity in a subset of neurons using a channelrhodopsin. We sparsely co-expressed ChrimsonR (*Klapoetke et al., 2014*) channelrhodopsin in neurons expressing rsCaMPARI and acquired images before and after illumination with marking light and pulsed 560 nm light to fully drive the channelrhodopsin (*Figure 2i–j* and *Figure 2—figure supplement 4*). A subset of neurons became much dimmer (*Figure 2i*, bottom panels), primarily corresponding to neurons that were positive for channelrhodopsin (*Figure 2i*, right panel; *Figure 2—figure supplement 4a*, bottom panels) and showed more than twofold contrast (*Figure 2j*). The same subset of neurons could be repeatedly erased and marked again (*Figure 2i*, cycles 2 and 3).

## Marking activity patterns in freely swimming zebrafish

To demonstrate the utility of rsCaMPARI for marking active neurons in vivo, we generated stable transgenic zebrafish (*Danio rerio*) expressing rsCaMPARI from a neuron-specific promoter (*Figure 3a*). We developed an imaging protocol for rsCaMPARI in the zebrafish brain using light sheet fluorescence microscopy for rapid acquisition and to minimize out-of-plane excitation light (*Figure 3b*). Zebrafish larvae (4–5 days post fertilization) experiencing a variety of stimuli were exposed to 10 s of marking light (400 mW/cm$^2$); 10 s exposure at similar light intensities gave good contrast under high- and low-calcium conditions (*Figure 1c* and *Figure 2c*) and was sufficient to observe distinct calcium activity patterns in the zebrafish brain using similar tools such as CaMPARI (*Fosque et al., 2015*). Light sheet z-stacks were acquired in the following order: (1) acquire the post-marking signal as a 'marked' stack, (2) reset to the bright on-state with erasing light, and (3) acquire the post-erased signal as a 'reference' stack. Comparison between the marked and reference stacks thus revealed the relative magnitude of rsCaMPARI photoswitching, which we quantitated and normalized as a ΔF/F metric. When the fish were anesthetized with the sodium channel blocker tricaine methanosulfonate (MS-222) to block brain activity during exposure to marking light, there was low and uniform labeling across the brain, consistent with a small amount of background photoswitching in the absence of calcium (*Figure 3c* and *Figure 3—figure supplement 1*). However, in freely swimming zebrafish, we observed neuron-specific labeling patterns, particularly in the forebrain, habenula, and hindbrain. Additional stimulation with cold or warm water, or the proconvulsant potassium channel blocker 4-aminopyridine (4-AP) also produced distinct and reproducible patterns of labeling. Generally, rsCaMPARI marked calcium activity patterns across brain areas similar to those observed previously with CaMPARI (*Fosque et al., 2015*; *Figure 3—figure supplement 2*), although some differences exist presumably due to variability across freely swimming fish. This variability may arise due to several sources, including differences in transgene expression patterns, differences in the zebrafish's perception of blue light versus violet light (*Guggiana-Nilo and Engert, 2016*; *Zimmermann et al., 2018*) used to mark calcium activity, image processing and analysis (comparison between rsCaMPARI ΔF/F images versus CaMPARI red images), and inherent variability from animal to animal and trial to trial that arise from the fish's current brain state leading up to and during the behavioral response.

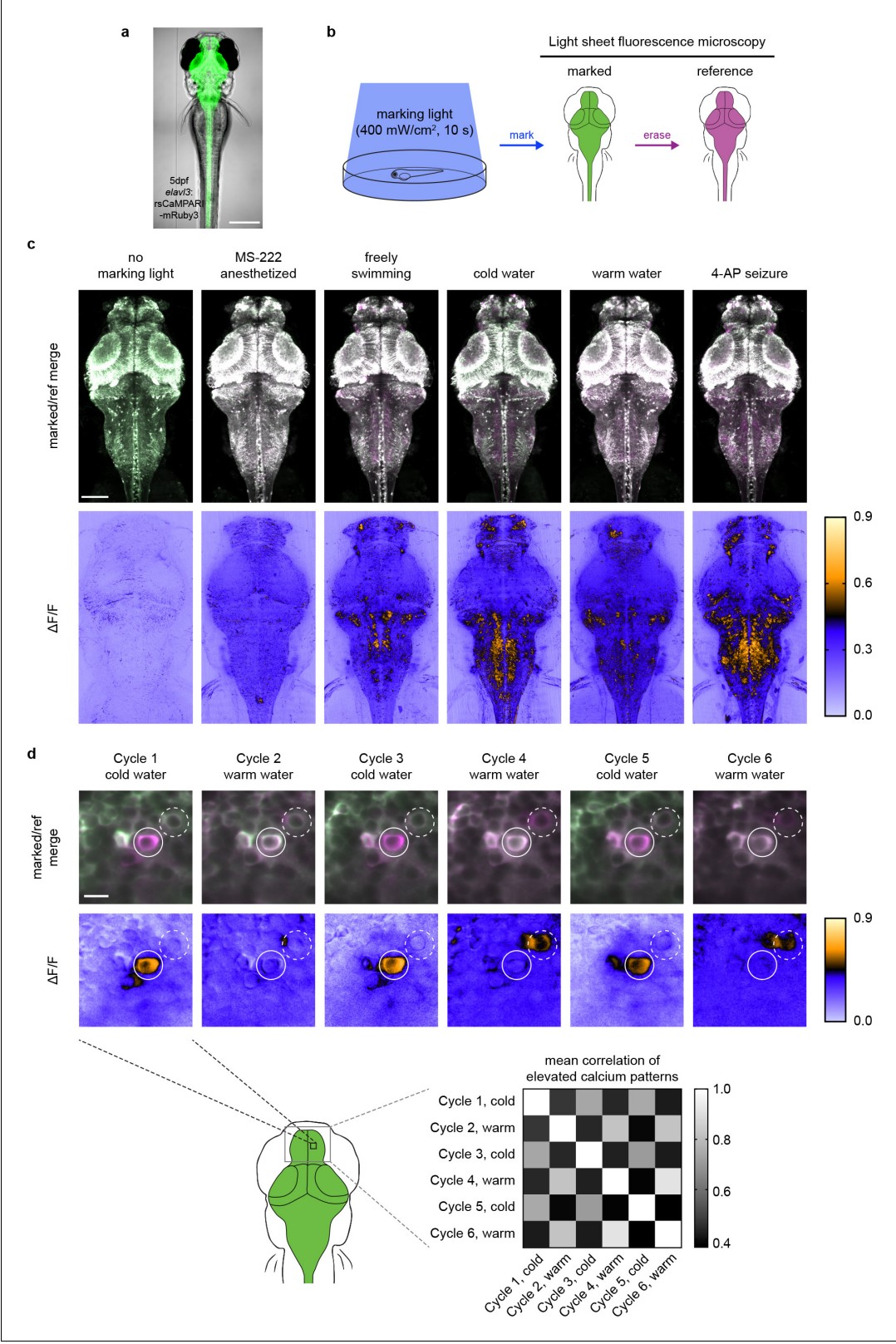

**Figure 3.** rsCaMPARI reversibly marks patterns of elevated calcium in the freely moving larval zebrafish. (**a**) rsCaMPARI expression in the Tg[*elavl3:* rsCaMPARI-mRuby3][if93] zebrafish. Scale bar is 300 μm. (**b**) Cartoon schematic of experimental setup and image acquisition. (**c**) Maximum intensity Z projections of the entire brain from zebrafish larvae (4 to 5 dpf) after exposure to different stimuli: no marking light, anesthetized with tricaine methanesulfonate (MS-222), freely swimming in system water, cold water (4°C), warm water (45°C), or 4-aminopyridine (4-AP). Top panels are merged

*Figure 3 continued on next page*

*Figure 3 continued*

reference and erased images, pseudo-colored green and magenta, respectively. Bottom panels are ΔF/F images. Scale bar is 100 μm. (d) Multiple cycles of rsCaMPARI marking in the same zebrafish (5 dpf) exposed to either cold or warm water. Top panels are individual Z slices from the pallium of the same fish brain illustrating one neuron (solid white circle) that was only labeled during cold stimulus and another neuron (dashed white circle) that was variably labeled during warm stimulus. Scale bar is 10 μm. Bottom-right panel is a mean correlation matrix comparing patterns of elevated calcium in the upper pallium and habenula across multiple marking cycles in the same fish.

The online version of this article includes the following source data and figure supplement(s) for figure 3:

**Source data 1.** Mean correlation matrix values of ΔF/F images across multiple cycles of rsCaMPARI marking in the same zebrafish pallium exposed to either cold or warm water.
**Figure supplement 1.** Replicates of rsCaMPARI labeling in zebrafish brain after exposure to different stimuli (same experiment as *Figure 3c*, different fish).
**Figure supplement 2.** Comparison of rsCaMPARI and CaMPARI labeling in zebrafish brain after exposure to different stimuli.
**Figure supplement 3.** Replicate of multiple rsCaMPARI labeling cycles in the same zebrafish (same experiment as *Figure 3d*, different fish).
**Figure supplement 3—source data 1.** Mean correlation matrix values of ΔF/F images across multiple cycles of rsCaMPARI marking in the same zebrafish pallium exposed to either cold or warm water.
**Figure supplement 4.** mRuby3 is poor for normalizing rsCaMPARI expression in larval zebrafish.
**Figure supplement 5.** Representative Z slices of rsCaMPARI labeling from different marking cycles in the same fish.

Finally, to demonstrate the ability of rsCaMPARI to reproducibly mark multiple distinct activity patterns in vivo, we performed multiple marking and erasing cycles in the same fish during alternating exposures to cold and warm water (*Figure 3d* and *Figure 3—figure supplement 3*). We observed stimulus-specific labeling across multiple marking cycles for neurons located in the zebrafish pallium. For example, over six marking cycles (alternating cold and warm water), specific neurons were strongly and repeatedly labeled during cold water exposures, but not during warm water stimulus (*Figure 3d*, upper panels). We also observed some variability across trials when using the same stimulus, presumably due to variability in the ongoing brain activity leading up to the stimulus and the variability in the perception to the stimulus experienced by freely swimming fish. Overall, the patterns of labeling across the pallium in response to either cold or warm water were specific to the stimulus: activity patterns in response to cold water showed higher mean correlation with other cold response patterns, and activity patterns in response to warm water showed higher mean correlation with other warm response patterns (*Figure 3d*, bottom-right panel). Taken together, these results demonstrate that rsCaMPARI is able to consistently capture patterns of neurons with elevated calcium from specific stimuli over multiple cycles in vivo.

## Discussion

The development of novel and improved calcium sensors continues to be immensely transformative for neuroscience, enabling experimental efforts toward decoding systems level dynamics and the mapping of functional circuits. Assignment of ensembles of neurons to specific functional circuits is greatly facilitated by comparisons between stimulus and control periods, or between repeated trials, which can be challenging for GCaMPs in freely moving animals over large spatial scales or impossible for CaMPARI due to its irreversible nature. In this work, we have developed a new type of erasable calcium marker called rsCaMPARI. Our experiments in zebrafish demonstrate how rsCaMPARI now enables marking of multiple patterns of cells with elevated calcium by allowing the probe to be reset by erasing light between trials. Thus, the erasable nature of rsCaMPARI now allows comparisons to be made between activity patterns across multiple trials from a variety of different stimuli, all within the same sample preparation.

We achieved erasable marking of neuronal activity by engineering reversibly switchable fluorescent proteins, a class of fluorescent probes that has been utilized for superresolution microscopy approaches but which has not yet been exploited to create functional activity reporters. Another possible future application of reversibly switchable activity reporters is their use with superresolution microscopy techniques like nonlinear structured illumination to visualize physiological signals within intracellular microdomains.

In summary, rsCaMPARI retains many of the advantages of a calcium marker such as bright fluorescence, high contrast, light gating, short integration windows (seconds), and stable integrated

signal, on top of being reusable. Additionally, rsCaMPARI can be used in a single (green) fluorescent color channel by comparing the post-marking light image to the post-erasing light image as we show in zebrafish, allowing the simultaneous use of complementary reporters or effectors in other color channels. We believe rsCaMPARI represents an important addition to the toolkit for marking active neuronal populations, and we expect it will enable functional circuit marking and mapping with higher accuracy than previously possible. Specifically, we envision rsCaMPARI being useful for single-neuron resolution mapping and comparison of activity patterns across the whole brain of freely swimming zebrafish during different behaviors or stimuli, as we have demonstrated here. We can also imagine similar experiments in mice. For example, marking and reading out activity patterns over regions of the cortex too large to image at single-neuron resolution with GCaMP, but doing so over numerous trials and behaviors in the same animal, which cannot be done with CaMPARI. This would enable better understanding of the interplay between cortical circuits at the single-neuron level.

# Materials and methods

## Key resources table

| Reagent type (species) or resource | Designation | Source or reference | Identifiers | Additional information |
|---|---|---|---|---|
| Gene (*Lobophyllia hemprichii*) | mEos3.1 | FPbase | FPbase:73GT1 | |
| Recombinant DNA reagent | pRSET_His-rsCaMPARI -mRuby3 (plasmid) | This paper | RRID:Addgene_120804 | Plasmid available at Addgene |
| Recombinant DNA reagent | pAAV-hsyn_NES-His -rsCaMPARI-mRuby3 (plasmid) | This paper | RRID:Addgene_120805 | Plasmid available at Addgene |
| Recombinant DNA reagent | pAAV-hsyn_NLS-His -rsCaMPARI-mRuby3 (plasmid) | This paper | RRID:Addgene_122092 | Plasmid available at Addgene |
| Recombinant DNA reagent | pTol2-elavl3_NES- rsCaMPARI-mRuby3 (plasmid) | This paper | RRID:Addgene_122129 | Plasmid available at Addgene |
| Recombinant DNA reagent | pAAV-hsyn_ChrimsonR -HaloTag (plasmid) | PMID:24509633 and PMID:18533659 | | |
| Strain, strain background (*Escherichia coli*) | T7 express | New England Biolabs | C2566 | Competent cells |
| Biological sample (*Rattus norvegicus*) | Primary rat hippocampal neurons | Janelia Research Campus | | Freshly isolated from *Rattus norvegicus* |
| Genetic reagent (*Danio rerio*) | Tg[*elavl3*:rsCa MPARI-mRuby3][jf93] | This paper | jf93Tg/+ | ZFIN ID: ZDB-FISH-191008–1 |
| Chemical compound, drug | Synaptic blockers | Tocris; PMID:24155972 | Cat# 0190, 0247, 1262, 0337 | |
| Chemical compound, drug | Janelia Fluor 635 dye, HaloTag ligand | PMID:28924668 | | Luke Lavis' lab |
| Chemical compound, drug | Tricaine methanesulfonate (MS-222) | Sigma | E10521 | |
| Chemical compound, drug | 4-Aminopyridine | Sigma | 275875 | |
| Software, algorithm | CMTK | PMID:12670015 | RRID:SCR_002234 | |

## Directed evolution of rsCaMPARI

CaMPARI2 in a pRSET plasmid (Life Technologies) was circularly permutated back to the original N- and C-termini of EosFP. The CaMPARI2 calmodulin and RS20 peptide were deleted to repair the original β-strand 7 of mEos3.1. H62 of the chromophore was mutated to L with a mutagenic primer

using the QuikChange method (Agilent) to produce a reversibly switchable mEos3.1 (rs-mEos3.1) construct variant. This construct contained the following mutations (derived from CaMPARI) relative to mEos3.1: V2 insert, F34Y, S39T, H62L, A69V, L93M, N102Y, N105S, C195T, L210I, and H213Y. A red fluorescent protein, mCherry (*Shaner et al., 2004*), was fused in frame at the C-terminus to normalize for expression and photoswitching. The plasmid was linearized by PCR such that calcium-binding domains could be inserted into β-strand 8 or 9. Calmodulin and RS20 were fused with a flexible $(GGS)_4$ amino acid linker in either the CaM-RS20 or RS20-CaM orientation for insertion as a single fragment using the Gibson assembly method (*Gibson et al., 2009*). The fragment was amplified with primers that contained (1) a variable region with 2 NNS codons plus 0 to 4 codons originating from EosFP in order to introduce linker composition and length diversity, and (2) a 5' 30 bp region for annealing to the linearized plasmid. Each β-strand library had a theoretical diversity size of $8 \times 10^6$. The Gibson assembly mixtures were diluted threefold in deionized water and transformed into T7 express *E. coli* (New England Biolabs) by electroporation. Transformed bacteria were plated on LB + ampicillin to isolate single colonies and a small number of clones were sequenced to confirm the desired library diversity.

Fluorescent colonies were manually picked and transferred to liquid growth media in 96-deep well plates and grown and harvested as previously described (*Fosque et al., 2015*). Two wells containing the rs-mEos3.1 construct without calcium-binding domain insertions and two wells containing a non-fluorescent variant, rs-Eos3.1(H62G, Y63G), were included in each plate as controls. To prepare lysates for screening, frozen cell pellets were resuspended in 800 µl lysis buffer (50 mM Tris, 150 mM NaCl, pH 8) containing 50% BPER (Thermo Fisher) and shaken at 30℃ for 30 min. Cell debris was pelleted by centrifugation and 95 µl of the cleared lysate was transferred to four separate 96-well microplates (*Figure 1—figure supplement 2*). Two of the plates were mixed with 5 µl calcium chloride solution to a final concentration of 0.5 mM. The other two plates were mixed with 5 µl EGTA solution to a final concentration of 1.0 mM. Green and red fluorescence intensities were measured on an Infinity M1000 fluorescence plate reader (Tecan). The plates were then illuminated with a 490 nm LED array (170 mW/cm$^2$, Luxeon) for 15 s before reading fluorescence intensities again. One of the original plates in high-calcium condition was swapped to high EGTA condition by adding 10 µl EGTA solution to a final concentration of 10 mM. Similarly, one of the original plates in high EGTA condition was swapped to high-calcium condition by adding 10 µl calcium chloride solution to a final concentration of 5 mM. Fluorescence intensities were measured again. Finally, the plates were illuminated with a 405 nm LED array (200 mW/cm$^2$, Loctite) for 10 s before reading final fluorescence intensities.

Library variants were selected based on four criteria: (1) contrast in green fluorescence following 490 nm light illumination $\pm$ Ca$^{2+}$, quantified as:

$$Average\ Contrast = \frac{\frac{F_G^{swap\ to\ EGTA}(swap)}{F_G^{stay\ in\ EGTA}(swap)} + \frac{F_G^{stay\ in\ Ca^{2+}}(swap)}{F_G^{swap\ to\ Ca^{2+}}(swap)}}{2}$$

(2) recovery of fluorescence intensity following 400 nm light illumination, (3) minimum fluorescence change due to calcium binding in the absence of light illumination (indicator behavior), quantified as:

$$Indicator\ Behavior = \frac{F_G^{EGTA}(pre) - F_G^{Ca^{2+}}(pre)}{F_G^{EGTA}(pre)}$$

and (4) green brightness. A total of 96 interesting library clones were selected for sequencing, resulting in 19 unique sequences. These variants were expressed and purified to measure relative brightness, indicator behavior, and photoswitching kinetics in the presence and absence of Ca$^{2+}$. One clone was renamed as rsCaMPARI ('rsCaMPARI-46' in *Figure 1—figure supplement 3*) and characterized further.

## rsCaMPARI expression and purification

rsCaMPARI protein was expressed with a N-terminal His$_6$ tag in T7 express *E. coli* (New England Biolabs) using a pRSET expression vector. The bacteria were grown in auto-induction media using the Studier method (*Studier, 2005*) for 36 hr at 30℃. Cells were pelleted by centrifugation, lysed in

Bacterial Protein Extraction Reagent (BPER)(Thermo Fisher), and the cleared lysate was loaded onto a column containing Ni-NTA agarose resin (QIAGEN). The column was washed with TBS (19.98 mM Tris, 136 mM NaCl, pH 7.4) + 10 mM imidazole before eluting with TBS + 200 mM imidazole. The eluted protein was loaded onto a Superdex 75 size-exclusion column (GE Healthcare) equilibrated with TBS and apparent monodispersity was confirmed. Fluorescent fractions were collected, combined, and stored at 4°C.

## In vitro analysis of purified protein

### Absorbance and emission spectra

Absorption spectra of rsCaMPARI were measured in TBS with 0.5 mM $CaCl_2$ or 1 mM EGTA on a NanoDrop One$^c$ UV-VIS spectrophotometer (Thermo) in cuvettes with 10 mm path length. Emission spectra were measured on an Infinity M1000 plate reader (Tecan) set to 495 nm excitation and scanning emission from 505 to 700 nm.

### Photophysical measurements

All the measurements were performed in 39 µM free calcium (+Ca) buffer (30 mM MOPS, 10 mM CaEGTA in 100 mM KCl, pH 7.2) or 0 µM free calcium (-Ca) buffer (30 mM MOPS, 10 mM EGTA in 100 mM KCl, pH 7.2). Absorbance measurements were performed using a UV-Vis spectrometer (Lamda 35, Perkin Elmer). Extinction coefficients were determined with the use of alkali denaturation method using extinction coefficient of denatured GFP as a reference ($\varepsilon$ = 44,000 $M^{-1}cm^{-1}$ at 447 nm). Quantum Yield measurements were performed using an integration sphere spectrometer (Quantaurus, Hamamatsu) for proteins in +Ca buffer.

### Two-photon measurements

The two-photon excitation spectra were performed as previously described (*Akerboom et al., 2012*). Protein solution of 1–2 µM concentration in +Ca or -Ca buffer was prepared and measured using an inverted microscope (IX81, Olympus) equipped with a 60x/1.2 NA water immersion objective (Olympus). Two-photon excitation was obtained using an 80 MHz Ti-Sapphire laser (Chameleon Ultra II, Coherent) for spectra from 710 nm to 1080 nm. Fluorescence collected by the objective was passed through a short pass filter (720SP, Semrock) and a band pass filter (550BP88, Semrock), and detected by a fiber-coupled Avalanche Photodiode (APD) (SPCM_AQRH-14, Perkin Elmer). The obtained two-photon excitation spectra were normalized for 1 µM concentration and further used to obtain the action cross-section spectra (AXS) with fluorescein as a reference (*Xu and Webb, 1996*; *Makarov et al., 2008*).

### Photoswitching rate measurements

To measure photoswitching rates ± calcium, purified rsCaMPARI in low (10 mM EGTA) or high (10 mM CaEGTA) calcium conditions buffered with 25 mM Tris, 100 mM KCl, pH 7.5 was illuminated with a 470 nm LED (Mightex) with different bandpass filters (parts: FB450-10, FB460-10, FB470-10, FB480-10, FB490-10, and FB500-10, Thorlabs; ET485/25x, Chroma). Output spectra were measured with a USB4000-UV-VIS Ocean Optics spectrometer. Light intensities were measured on a power meter (Coherent #1098580) using a silicon photodiode (Coherent #1098313). Green fluorescence was measured on a plate reader (Tecan) at various timepoints and a single exponential rate was fitted (Prism, Graphpad) to the change in green fluorescence. The optimal wavelengths for rsCaMPARI off-switching (470–490 nm) are defined as 'marking light' and the optimal wavelengths for rsCaMPARI on-switching (390–410 nm) are defined as 'erasing light'.

### $K_d$ measurements

Solutions of EGTA-buffered free $Ca^{2+}$ were prepared using a pH-titration method as previously described (*Tsien and Pozzan, 1989*) in 25 mM Tris, 100 mM KCl, pH 7.5. Free $Ca^{2+}$ concentration was calculated using the following parameters for EGTA solutions with 0.1 M ionic strength at 25 °C (*Martell et al., 2004*): log $K_{Ca}$ = 10.86, $pK_1$ = 9.51, $pK_2$ = 8.90. Note that $pK_1$ and $pK_2$ are adjusted 0.11 higher for 0.1 M ionic strength as explained by *Tsien and Pozzan, 1989*. Therefore, the effective $K_d(Ca)$ for EGTA is calculated as follows:

$$K_d(\text{Ca}) = \frac{[\text{Ca}^{2+}]\left[\text{EGTA}^{4-} + \text{HEGTA}^{3-} + \text{H}_2\text{EGTA}^{2-}\right]}{\left[\text{CaEGTA}^{2-}\right]}$$

$$= \frac{[\text{Ca}^{2+}]\left[\text{EGTA}^{4-}\right]\left[1 + 10^{(\text{p}K_1-\text{pH})} + 10^{(\text{p}K_2+pK_1-2\text{pH})}\right]}{\left[\text{CaEGTA}^{2-}\right]}$$

$$= \frac{\left[1 + 10^{(\text{p}K_1-\text{pH})} + 10^{(\text{p}K_2+pK_1-2\text{pH})}\right]}{K_{\text{Ca}}}$$

$$= \frac{\left[1 + 10^{(9.51-7.5)} + 10^{(8.90+9.51-2(7.5))}\right]}{10^{10.86}}$$

$$= 3.69 \times 10^{-8}\text{M}$$

Ca-EGTA solutions over a range of free $\text{Ca}^{2+}$ concentrations from 5.5 nM to 19 µM were prepared by mixing various volumes of a 10 mM Ca-EGTA solution with a 10 mM EGTA solution. To measure apparent affinity of rsCaMPARI for calcium ions, 2 µl of protein solution (~50 µM) was mixed with 98 µl of different Ca-EGTA solutions and the fluorescence intensity (Ex. 500 nm, Em. 515 nm) was measured on an Infinity M1000 fluorescence plate reader (Tecan). The data were fit (Prism, Graphpad) to a binding curve of the form:

$$F = F_{max} \times \left(1 - \frac{x^h}{K_d^h + x^h}\right) + F_{min}$$

where

$$
\begin{array}{ll}
F & = \text{fluorescence signal (AU)} \\
F_{max} & = \text{maximum fluorescence signal (AU)} \\
F_{min} & = \text{minimum fluorescence signal (AU)} \\
x & = \text{concentration of free } \text{Ca}^{2+} \text{ (M)} \\
K_d & = \text{dissociation constant (M)} \\
h & = \text{Hill coefficient}
\end{array}
$$

$F_{max}$ was normalized to one to plot the relative fluorescence intensity. To measure rsCaMPARI off-switching rate as a function of free $\text{Ca}^{2+}$ concentration, protein in Ca-EGTA solutions were prepared as described above and illuminated with various intervals of marking light (200 mW/cm$^2$). The fluorescence intensity was measured at each timepoint to fit a single exponential. The extrapolated kinetic rate $k$ was then fitted to a binding curve of the form:

$$k = k_{max} \times \frac{x^h}{K_d^h + x^h} + k_{min}$$

where

$$
\begin{array}{ll}
k & = \text{kinetic rate (s}^{-1}) \\
k_{max} & = \text{maximum kinetic rate (s}^{-1}) \\
k_{min} & = \text{minimum kinetic rate (s}^{-1}) \\
x & = \text{concentration of free } \text{Ca}^{2+} \text{ (M)} \\
K_d & = \text{dissociation constant (M)} \\
h & = \text{Hill coefficient}
\end{array}
$$

$k_{min}$ was normalized to one to plot the relative off-switching rate.

## rsCaMPARI experiments in dissociated primary neuron cultures

All procedures involving animals were conducted in accordance with protocols approved by the Howard Hughes Medical Institute (HHMI) Janelia Research Campus Institutional Animal Care and Use Committee and Institutional Biosafety Committee (protocol 18–168). Hippocampal neurons extracted from P0 to 1 Sprague-Dawley rat pups were transfected with plasmids (see below) or plated directly without transfection into either 24-well glass-bottom plates (Mattek, #1.5 coverslip) for field electrode and channelrhodopsin experiments or 25 mm ultra-clean cover glasses (Sigma) for electrophysiology experiments. All glass surfaces were coated with poly-D-lysine prior to plating. Neurons were cultured in NbActiv4 medium (BrainBits) at 37°C with 5% $CO_2$ in a humidified atmosphere. All imaging measurements were performed in imaging buffer containing 10 mM HEPES, 145

mM NaCl, 2.5 mM KCl, 10 mM glucose, 2 mM $CaCl_2$, 1 mM $MgCl_2$, pH 7.4 supplemented with synaptic blockers (10 µM CNQX, 10 µM CPP, 10 µM GABAZINE, and 1 mM MCPG; Tocris Bioscience) to block ionotropic glutamate, GABA, and metabotropic glutamate receptors (*Wardill et al., 2013*).

All images were acquired on an inverted Nikon Eclipse Ti2 microscope equipped with a SPECTRA X light engine (Lumencor), an AURA II light engine (Lumencor), and an ORCA-Flash 4.0 sCMOS camera (Hamamatsu). The SPECTRA X light engine was equipped with 485/25 and 395/25 nm excitation filters to produce marking and erasing light, respectively; and 550/15 and 640/30 nm excitation filters for exciting mRuby3 and JF635 dye, respectively. A quad bandpass filter (set number: 89000, Chroma) was used along with 525/36, 605/52, and 705/72 nm emission filters (Chroma) to image rsCaMPARI, mRuby3, and JF635 dye, respectively. A 550SP dichroic mirror (Thorlabs) was used to filter 560 nm light from the AURA II light engine for photostimulation of the ChrimsonR channelrhodopsin. Light powers were measured using a power meter (Thorlabs PM100A) with a Si photodiode (Thorlabs S120C or S170C). All image analysis and quantification were performed using Fiji software (*Schindelin et al., 2012*). Please see *Source code 1* for details on image processing and corresponding macro scripts.

## Characterization of rsCaMPARI in primary neuron cultures stimulated with a field electrode

Dissociated neurons were plated directly without transfection. Three days after plating, the neurons were infected with AAV2/1 virus encoding rsCaMPARI-mRuby3 under control of the hsyn1 promoter. Imaging was performed 5–7 days after infection. The neurons were exposed to 2–20 s of continuous marking light (224 mW/cm$^2$) through a 10x/0.45 NA objective (Nikon) for simultaneous photoswitching and imaging (10 Hz acquisition) of rsCaMPARI during one marking cycle. Each marking cycle was erased with 3 s of erasing light (224 mW/cm$^2$ power). Field stimulations (5–80 Hz, 1 ms) were produced using a custom-built field electrode controlled by a high current isolator (A385, World Precision Instruments) set to 90 mA. Stimulations were controlled using an Arduino Uno board and synchronized with light sources in Nikon Elements software. To plot fluorescence time-course traces, rsCaMPARI green signals are normalized to the first image representing the pre-marked green signal. A ΔF/F metric was calculated to contrast photoswitching differences between stimulated and non-stimulated conditions, where ΔF/F is defined as:

$$\Delta\text{F/F} = \frac{F_{pre-marked} - F_{post-marked}}{F_{pre-marked}}$$

For analysis using mRuby3 red signal for normalization, rsCaMPARI green signals are normalized to initial pre-marked red signal.

For characterizing rsCaMPARI spontaneous recovery in the dark, neurons that received prior exposure to 20 s of marking light ± field stimulation from one marking cycle were incubated in the dark in a stage top incubator (Tokai Hit) set to 37°C. Snapshots of rsCaMPARI were acquired every 5 min (224 mW/cm$^2$, 100 ms). In between each snapshot, 160 stimulations (80 Hz, 1 ms) were delivered using the field electrode. After 60 min, rsCaMPARI was erased with ~3 s of erasing light (405 nm, 224 mW/cm$^2$ power) before a final snapshot of rsCaMPARI was acquired. For characterizing rsCaMPARI photofatigue across multiple marking cycles, neurons were continually marked and erased with 20 s of marking light (224 mW/cm$^2$) and 3 s of erasing light (224 mW/cm$^2$ power), respectively. Field stimulations (3 × 160 stims, 80 Hz, 1 ms) were delivered during marking light illumination on every odd cycle.

## Simultaneous electrophysiology and fluorescence imaging in primary neuron culture

Dissociated neurons were transfected with a pAAV plasmid containing rsCaMPARI-mRuby3 under control of the hsyn1 promoter by electroporation (Lonza, P3 Primary Cell 4D-Nucleofector X kit) according to the manufacturer's instruction. Nine days after plating, the neurons were exchanged to imaging buffer containing synaptic blockers for imaging. Within a field of view containing multiple neurons expressing rsCaMPARI, a single neuron was patched and the entire field of view was exposed to 15 s of marking light (150 mW/cm$^2$) through a 40x/1.3 NA oil objective (Nikon) for simultaneous photoswitching and imaging (10 Hz acquisition) of rsCaMPARI during one marking cycle.

Each marking cycle was erased with ~3 s of erasing light (486 mW/cm$^2$). To plot fluorescence time-course traces, rsCaMPARI green signals are normalized to the first image representing the pre-marked green signal. $\Delta F/F$ was calculated to contrast photoswitching differences between current injection and non-patched conditions.

Whole-cell patch clamp recordings were performed with filamented glass micropipettes (Sutter instruments) pulled to a tip resistance of 10–12 M$\Omega$. The internal solution in the pipette contained (in mM): 130 potassium methanesulfonate, 10 HEPES, 5 NaCl, 1 MgCl2, 1 Mg-ATP, 0.4 Na-GTP, 14 Tris-phosphocreatine, adjusted to pH 7.3 with KOH, and adjusted to 300 mOsm with sucrose. Pipettes were positioned using a MPC200 manipulator (Sutter instruments) and current clamp traces were recorded using an EPC800 amplifier (HEKA) and digitized using a National Instruments PCIe-6353 acquisition board. To generate action potentials, current was injected to induce spike trains (3 × 20–200 pA for 1 s) and voltage was monitored. WaveSurfer software (https://wavesurfer.janelia.org/) was used to control the amplifier, camera, light source, and record voltage and current traces.

## rsCaMPARI experiments in primary neurons with a subset driven by a channelrhodopsin

Dissociated neurons were transfected with a pAAV plasmid containing a ChrimsonR (*Klapoetke et al., 2014*) and HaloTag (*Los et al., 2008*) fusion protein under control of the hsyn1 promoter by electroporation (Lonza, P3 Primary Cell 4D-Nucleofector X kit) according to the manufacturer's instruction and mixed 2:1 with non-transfected cells before plating. 6 days after plating, the neurons were infected with AAV2/1 virus encoding rsCaMPARI-mRuby3 under control of hsyn1. Imaging was performed 7 days after infection. To label neurons expressing ChrimsonR-HaloTag, cultures were incubated with 100 nM JF635-HaloTag (*Grimm et al., 2017*) ligand for 30 min. Neurons were then washed in imaging buffer three times and the buffer was then replaced with imaging buffer containing synaptic blockers. The neurons were exposed to 10 s of continuous marking light (285 mW/cm$^2$) through a 10x/0.45 NA objective (Nikon) for simultaneous photoswitching and imaging (10 Hz acquisition) of rsCaMPARI during one marking cycle. 560 nm light (59 mW/cm$^2$) was also pulsed (10 ms pulses at 10 Hz) through the objective during the entire marking light illumination to fully drive the channelrhodopsin. Each marking cycle was erased with ~3 s of erasing light (153 mW/cm$^2$ power). Classification of neurons expressing channelrhodopsin was done using a threshold of the JF635 mean fluorescence. $\Delta F/F$ was calculated to contrast photoswitching differences +ChR and -ChR conditions. For analysis using mRuby3 red signal for normalization, rsCaMPARI green signals are normalized to initial pre-marked red signal.

## rsCaMPARI experiments in larval zebrafish (*Danio rerio*)

All zebrafish experiments were conducted in accordance with the animal research guidelines from the National Institutes of Health and were approved by the Institutional Animal Care and Use Committee and Institutional Biosafety Committee of Janelia Research Campus (protocol 18–173).

To generate the Tg[*elavl3*:rsCaMPARI-mRuby3]$^{jf93}$ line, 1–2 cell embryos from *casper* background zebrafish were injected with a Tol2 vector containing rsCaMPARI-mRuby3 under control of the *elavl3* pan-neuronal promoter. Potential founders were screened for bright green and red fluorescence in the brain and later crossed with *casper* background zebrafish to screen for progeny (F1 generation) exhibiting pan-neuronal rsCaMPARI-mRuby3 expression in the central nervous system (CNS). The F1 generation fish were later incrossed and 4–5 days post-fertilization (dpf) larvae exhibiting bright green fluorescence in the CNS were used for experiments.

We observed poor correlation between the rsCaMPARI green signal and the mRuby3 red signal in the larvae (*Figure 3—figure supplement 4*), presumably because the larval zebrafish brain is rapidly developing and the maturation time of the mRuby3 chromophore is slow (*Balleza et al., 2018*) compared to rsCaMPARI. Therefore, we caution against using mRuby3 as an expression normalization tag in larval zebrafish. However, the mRuby3 tag was useful for initially locating and positioning the zebrafish brain within the field of view for imaging without using blue light, which would induce further rsCaMPARI photoswitching.

Zebrafish in system water were illuminated for 10 s with marking light (470 nm Mightex LED fitted with a Chroma 485/25x filter, 400 mW/cm$^2$) under various stimulus conditions, including freely swimming in system water, anesthetized in 0.24 mg/mL tricaine methanesulfonate (MS-222, Sigma), cold

water (4°C), warm water (45°C), and following 15–30 min exposure to 800 µM 4-aminopyridine (4-AP, Sigma). Following marking light exposure, the fish was anesthetized in 0.24 mg/mL MS-222 and immobilized with 2% agarose in a glass capillary tube (size 2, Zeiss) for imaging. For fish undergoing multiple cycles of marking and erasing, the fish was transferred to fresh system water and carefully removed from the agarose. The fish was allowed to recover to freely swimming behavior (typically within 15 min) and a brief 3 s illumination with erasing light (405 nm Loctite LED, 200 mW/cm$^2$) was used to reset the sensor before another marking cycle began.

Larval zebrafish brains were imaged on a Zeiss Lightsheet Z.1 microscope equipped with 50 mW 405, 488, and 561 nm laser lines and a pco.edge 5.5 sCMOS camera (PCO). Images were acquired using either a 10x/0.5 NA (for whole brain) or 20x/1.0 NA (for forebrain) water immersion objective (Zeiss) for detection, and two 10x/0.2 NA optics (Zeiss) for dual-side illumination. The sample chamber was filled with system water containing 0.24 mg/mL MS-222 and the solidified agarose was partially extruded from the capillary tube to position the fish within the sample chamber. A 585 nm longpass emission filter was used to visualize mRuby3 fluorescence for positioning and a 505–545 nm bandpass emission filter was used to image rsCaMPARI fluorescence. Each Z slice was acquired in dual-side illumination mode with pivot scan using 1% 488 nm excitation and 100 ms exposure time. Two Z stacks (3 µm steps for whole brain or 2 µm steps for forebrain) were acquired using continuous drive mode in the following order: (1) acquire first stack as the 'marked' dataset (green), (2) erase the stack with 5% 405 nm light, and (3) acquire second stack as the 'reference' dataset (pseudo-colored magenta). Total time to acquire both marked and reference stacks was ~1–2 min.

Acquired images were dual-side fused in Zeiss ZEN software. Motion drift between marked and reference stacks were corrected using affine alignment tools in Computational Morphometry ToolKit (*Rohlfing and Maurer, 2003*) (CMTK) software (https://www.nitrc.org/projects/cmtk/) with the following parameters: exploration 26, accuracy 0.4, and dofs 12. Composite marked and reference images and ΔF/F images using the ICA lookup table were created in Fiji (*Schindelin et al., 2012*). Here, ΔF/F is defined as:

$$\Delta \mathrm{F}/\mathrm{F} = \frac{F_{reference} - F_{marked}}{F_{reference}}$$

To compare rsCaMPARI labeling with CaMPARI labeling, a CaMPARI dataset from *Fosque et al., 2015* was used. CaMPARI confocal images of larval zebrafish brain exposed to MS-222, cold water, warm water, 4-AP, and freely swimming conditions were aligned to the rsCaMPARI images. The CaMPARI green image stack was first aligned with a rsCaMPARI reference image stack using an affine alignment in CMTK with the following parameters: exploration 26, accuracy 0.4, and dofs 6. The CaMPARI red image stack was then reformatted in CMTK to align with the respective CaMPARI green image stack. Finally, the CaMPARI red images were pseudo-colored using the ICA lookup table in Fiji and presented with the corresponding rsCaMPARI ΔF/F images from the same stimulus condition and Z slice.

To compute mean correlation coefficients between trials in the same fish, reference image stacks of the forebrain were first aligned to each other with an initial affine alignment in CMTK with the following parameters: exploration 26, accuracy 0.4, and dofs 12; the images were then further aligned with a restrained warp in CMTK with the following parameters: exploration 26, coarsest 4, grid-spacing 90, refine 3, accuracy 0.4, and jacobian-weight 0.05. Marked image stacks were reformatted in CMTK to align with its respective reference stack and ΔF/F images were calculated. The resulting ΔF/F image stacks were cropped to remove sub-pallium regions where excitation light was scattered by pigmentation in the eyes of the zebrafish, and brain tissue with rsCaMPARI expression was selected by a 0.25 ΔF/F threshold. Representative slices are shown in *Figure 3—figure supplement 5*. Mean correlations between image stacks were calculated using cosine similarity across all slices using a custom macro in Fiji according to the following formula:

$$mean\ correlation = \frac{\sum_{i=1}^{ZMN} A_i B_i}{\sqrt{\sum_{i=1}^{ZMN} A_i^2 \sum_{i=1}^{ZMN} B_i^2}}$$

where A and B are image stacks with Z slices, and each slice has dimension M x N pixels. Please see *Source code 1* for full macro script.

## Practical considerations for the use of rsCaMPARI

### Photoswitching light source, intensity, and duration

rsCaMPARI is a reversibly switchable fluorescent protein that requires a source of 470–490 nm blue marking light and ∼400 nm violet erasing light. We observed that the off-switching rate contrast was dependent on the wavelength of blue light used and that using lower wavelengths resulted in better off-switching rate contrast (∼10 fold at 500 nm to ∼25 fold at 450 nm) (see *Figure 1—figure supplement 8a–c*). However, when exposed to wavelengths < 470 nm, rsCaMPARI never reached the fully off-switched state and instead settled at an intermediate level of fluorescence (see *Figure 1—figure supplement 8b*). We attributed this behavior to spectral overlap of the switching light with the broad 392 nm peak that drives rsCaMPARI on-switching. Indeed, when starting from the fully off-switched state, 450–470 nm light on-switches the protein (see *Figure 1—figure supplement 8d*). Therefore, the observed intermediate resting fluorescence is due to 450–470 nm light driving both on- and off-switching to achieve an intermediate equilibrium state. For the marking light, we recommend that wavelengths between 470 nm and 490 nm should be used with rsCaMPARI to maintain high off-switching rate contrast while avoiding on-switching, and light that falls outside these wavelengths should be filtered out. The light source wavelengths should be verified, and it may be helpful to optimize light sources as we demonstrated in *Figure 1—figure supplement 8*.

The marking light intensity and duration is dependent on experimental parameters, which include how long the behavior of interest occurs and the level of underlying calcium activity. For rapid photoswitching, a bright light source is required. The photoswitching rate is linear with respect to light intensity (see *Figure 1—figure supplement 9*), therefore the light intensity may be adjusted to match the desired light duration. Generating a photoswitching curve, for example see *Figure 1c* and *Figure 1—figure supplement 8b*, may be helpful for determining the appropriate light intensity and duration using a particular light source. In this work, we have successfully used marking light intensities from ∼200 to 400 mW/cm$^2$ over a duration of 10–20 s to image calcium activity in neuron culture and larval zebrafish brain. Erasing light requirements are less stringent, because the erasing light does not need to coincide with the behavior of interest and on-switching is much more efficient than off-switching. We have successfully used erasing light intensity ∼200 mW/cm$^2$ over a duration of a few seconds.

### Sample handling and imaging rsCaMPARI signal

Care should be taken to not expose the sample to spurious and bright light in the blue and violet range, especially between marking light exposure and image acquisition. rsCaMPARI signal is stable in the dark (see *Figure 2d*) and we generally handle samples under dim ambient light conditions. If the sample is exposed to spurious light that compromises the marked signal, the erasable nature of rsCaMPARI allows the tool to be easily reset for another marking cycle. We consider it good practice to initialize rsCaMPARI before each marking cycle with a short exposure of erasing light to ensure the tool is completely reset.

A key consideration for imaging rsCaMPARI is that the marking light is the same wavelength as the excitation light. For tissue volumes this means that the sample is very sensitive to out-of-plane excitation light and methods such as confocal microscopy are not suitable for imaging rsCaMPARI. We have found success with imaging rsCaMPARI in tissue volumes using light sheet fluorescence microscopy, which has good sectioning capabilities to keep the excitation light in-plane and allows rapid acquisition across the entire tissue volume. The sensitivity of the sample to excitation light also means that initial positioning of the sample within the sample chamber should not rely on rsCaMPARI. A fiducial marker in a separate color channel is ideal for positioning the sample and we have successfully used mRuby3 for this purpose.

Similar to CaMPARI, rsCaMPARI fluorescence is dependent on calcium (see *Figure 1f*) and should therefore be imaged under uniformly low calcium conditions. For cultured neurons, we imaged in the presence of synaptic blockers to prevent spontaneous activity; and for the zebrafish, we imaged under anesthetized conditions using the sodium channel blocker tricaine methanesulfonate (MS-222) to prevent action potential firing.

## Quantification of rsCaMPARI photoswitching

To normalize the magnitude of rsCaMPARI fluorescence change with protein expression, the post-marked green intensities must be quantified in relation to a reference image. This may be achieved in a couple ways. The first method is a ΔF/F metric, which normalizes the post-marked intensity change with either the pre-marked green image or an erased green image. For motionless samples, such as neuron culture, acquiring the pre-marked or erased image is relatively simple. The marking light is the same wavelength as the excitation light for green image acquisition, therefore the pre-marked image can be acquired concurrently during the first frame of marking light illumination. The erased image can be acquired after a short exposure to erasing light illumination. For moving samples, such as the zebrafish, acquiring the pre-marked image is more challenging, but the erased image can be easily acquired after the post-marked image acquisition while the sample is still immobilized on the imaging stage. The second method uses a separate channel to image a fiducial marker that is fused to rsCaMPARI. We demonstrated this method using mRuby3 in neuron culture (see *Figure 2—figure supplement 1* and *Figure 2—figure supplement 4*) and showed there is good correlation between mRuby3 normalization and ΔF/F (see *Figure 2—figure supplement 4c*). The choice of either method depends on experimental need and has certain advantages and disadvantages. For example, ΔF/F requires only a single-color channel, but may be more difficult to register due to separation in time between each image acquisition. A fiducial marker like mRuby3 can be acquired concurrently with rsCaMPARI, but may not correlate well with rsCaMPARI in a particular experimental system, such as the rapidly developing larval zebrafish brain (*Figure 3—figure supplement 4*).

# Acknowledgements

We thank members of the Janelia Experimental Technology (jET), Media Prep, Molecular Biology, Viral Tools, Vivarium, and Light Microscopy cores. Specifically, we thank Damien Alcor, Michael DeSantis, Benjamin Foster, Vasily Goncharov, Cameron Loper, Igor Negrashov, Kimberly Ritola, Jared Rouchard, Steven Sawtelle, Jordan Towne, Deepika Walpita, and Xiaorong Zhang. We thank Kaspar Podgorski for comments on the project and manuscript.

# Additional information

### Competing interests

Fern Sha, Eric R Schreiter: is listed as an inventor on a patent application describing reversibly switchable neuronal activity markers. The other authors declare that no competing interests exist.

### Funding

| Funder | Author |
|---|---|
| Howard Hughes Medical Institute | Eric R Schreiter |

The funders had no role in study design, data collection and interpretation, or the decision to submit the work for publication.

### Author contributions

Fern Sha, Conceptualization, Data curation, Formal analysis, Investigation, Methodology, Writing - original draft, Project administration, Writing - review and editing; Ahmed S Abdelfattah, Ronak Patel, Investigation, Methodology; Eric R Schreiter, Conceptualization, Supervision, Funding acquisition, Methodology, Writing - original draft, Writing - review and editing

### Author ORCIDs

Fern Sha ⬤ http://orcid.org/0000-0002-7110-2544
Eric R Schreiter ⬤ https://orcid.org/0000-0002-2864-7469

### Ethics

Animal experimentation: All procedures involving animals were conducted in accordance with protocols approved by the Howard Hughes Medical Institute (HHMI) Janelia Research Campus Institutional Animal Care and Use Committee and Institutional Biosafety Committee (protocols 18-168 and 18-173).

### Decision letter and Author response

Decision letter https://doi.org/10.7554/eLife.57249.sa1
Author response https://doi.org/10.7554/eLife.57249.sa2

## Additional files

### Supplementary files

• Source code 1. Source code for all Fiji macro scripts used in image analysis are provided.

• Transparent reporting form

### Data availability

DNA constructs for pRSET_His-rsCaMPARI-mRuby3, pAAV-hsyn_NES-His-rsCaMPARI-mRuby3, pAAV-hsyn_NLS-His-rsCaMPARI-mRuby3, and pTol2-elavl3_NES-rsCaMPARI-mRuby3 are available via Addgene (http://www.addgene.org #120804, #120805, #122092, and #122129, respectively). Tg [elavl3:rsCaMPARI-mRuby3]jf93 transgenic zebrafish are deposited to the Zebrafish International Resource Center (https://zebrafish.org). Source data and data analysis algorithms are provided.

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
