## [Decision Letter]

Thank you for submitting your article "rsCaMPARI: an erasable marker of neuronal activity" for consideration by *eLife*. Your article has been reviewed by two peer reviewers, and the evaluation has been overseen by a Reviewing Editor and Richard Aldrich as the Senior Editor. The following individual involved in review of your submission has agreed to reveal their identity: Amy E Palmer (Reviewer #1).

The reviewers have discussed the reviews with one another and the Reviewing Editor has drafted this decision to help you prepare a revised submission.

Summary:

In this paper, Zha and colleagues introduce a new reversibly switchable calcium sensor, rsCAMPARI, that can be used to repetitively mark active neurons in vitro and in vivo. Calcium binding to the probe accelerates its dimming in response to blue light, identifying neurons that were active during the time of illumination, and violet light resets the probe for additional measurements. A major potential advantage of this probe over existing methods is that it may allow large-scale mapping of neural circuits in the same animal over time, enabling comparisons of variations of neural circuitry or response patterns within a single or among multiple individuals. The properties of rsCAMPARI are well-suited to delineate active neuronal networks; its response is graded with activity, the marked state is stable enough to permit delayed microscopic analysis up to an hour after marking, and the probe can distinguish activated neurons over 10 cycles. The authors illustrate its practical application in primary neuronal cultures upon field stimulation and in freely-swimming zebrafish exposed to warm and cold water.

Essential revisions:

Both reviewers were impressed with the innovative design of rsCAMPARI and the quality of the results, and agreed that it is potentially a powerful tool, particularly for tracking spatial patterns of activity over multiple trials in freely behaving animals. However, they noted several essential issues that will need to be addressed, as detailed below. Given the difficulty of conducting lab experiments during the current pandemic, the number of required new experiments has been held to a minimum, and most of the changes can be made through revisions to the text.

1) A major concern is the lack of evidence showing that rsCAMPARI will enable live-animal studies that could not be carried out with existing tools. This bears on the significance and utility of rsCAMPARI and is mentioned only briefly in the Introduction and Discussion (first paragraph). To make the rationale for its use clearer for a broader audience, the limitations of current methods need to be described more fully with citations in the Introduction and/or Discussion. It will also be important to discuss the particular kinds of questions to which rsCAMPARI could be applied, to underscore the potential power of this new tool. Without stronger support, the last sentence of the Discussion ("…we expect it will enable functional circuit marking and mapping with higher accuracy than previously possible.") sounds more like a hope than a likely outcome.

2) A more complete description of rsCAMPARI performance and technical limitations is needed, particularly in comparison to existing probes. A small and variable number of neurons appear active in the experiment of Figure 3D, and the question is whether the low number and high variability is a natural result of the biology or of the limited S/N properties of the probe. The reviewers suggest comparing the response of rsCAMPARI with that of GCaMP (or CAMPARI) in immobilized zebrafish subjected to cold/warm water. Does rsCAMPARI indicate all cells that fire? This was seen as a critical issue that would affect the choice of indicator for a variety of experiments and would determine whether rsCAMPARI is suitable for reconstructing a complete neuronal circuit.

3) A more systematic discussion is needed of the experimental parameters (marking time, observation window, illumination intensity) that define the usability window of rsCAMPARI. What factors led to the choice of 10 s marking time? Would the probe be useful for much shorter or longer integration times, and what considerations would affect this? How should the combination of illumination intensity and marking time be optimized?

---

## [Author Response]

Essential revisions:Both reviewers were impressed with the innovative design of rsCAMPARI and the quality of the results, and agreed that it is potentially a powerful tool, particularly for tracking spatial patterns of activity over multiple trials in freely behaving animals. However, they noted several essential issues that will need to be addressed, as detailed below. Given the difficulty of conducting lab experiments during the current pandemic, the number of required new experiments has been held to a minimum, and most of the changes can be made through revisions to the text.1) A major concern is the lack of evidence showing that rsCAMPARI will enable live-animal studies that could not be carried out with existing tools. This bears on the significance and utility of rsCAMPARI and is mentioned only briefly in the Introduction and Discussion (first paragraph). To make the rationale for its use clearer for a broader audience, the limitations of current methods need to be described more fully with citations in the Introduction and/or Discussion. It will also be important to discuss the particular kinds of questions to which rsCAMPARI could be applied, to underscore the potential power of this new tool. Without stronger support, the last sentence of the Discussion ("…we expect it will enable functional circuit marking and mapping with higher accuracy than previously possible.") sounds more like a hope than a likely outcome.

We have added text to the Introduction discussing existing technologies for marking active neuron populations and their limitations. Additionally, we have added text to the Discussion about possible applications of rsCaMPARI.

2) A more complete description of rsCAMPARI performance and technical limitations is needed, particularly in comparison to existing probes. A small and variable number of neurons appear active in the experiment of Figure 3D, and the question is whether the low number and high variability is a natural result of the biology or of the limited S/N properties of the probe. The reviewers suggest comparing the response of rsCAMPARI with that of GCaMP (or CAMPARI) in immobilized zebrafish subjected to cold/warm water. Does rsCAMPARI indicate all cells that fire? This was seen as a critical issue that would affect the choice of indicator for a variety of experiments and would determine whether rsCAMPARI is suitable for reconstructing a complete neuronal circuit.

Since we have included a quantitative characterization of rsCaMPARI function in Table 1, it is possible to directly compare the photophysical properties of rsCaMPARI (brightness, calcium affinity, contrast) with those of existing published probes such as CaMPARI or GCaMP. Additionally, responses in primary neuron cultures to controlled stimuli like those presented in panel 2B can also be quantitively compared between rsCaMPARI and other sensors. We believe that this is the best way for potential users to determine how rsCaMPARI may perform in their system relative to more familiar tools.

in vivo comparison of different tools is difficult, but we have added Figure 3—figure supplement 2 showing a visual comparison of rsCaMPARI and CaMPARI responses to several stimuli in transgenic larval zebrafish. There are spatial patterns of activity observed across the brain with rsCaMPARI that are generally similar to those observed with CaMPARI, but some differences exist due to differences in transgene expression patterns in these different transgenic animals, differences in the way the signal is marked (blue vs. violet light) and processed (comparison of marked and erased rsCaMPARI images vs. a single image of CaMPARI red or red/green), and variability from animal to animal and trial to trial that result from the animals current brain state and exactly how they experience the stimulus in a freely swimming behavioral setup.

3) A more systematic discussion is needed of the experimental parameters (marking time, observation window, illumination intensity) that define the usability window of rsCAMPARI. What factors led to the choice of 10 s marking time? Would the probe be useful for much shorter or longer integration times, and what considerations would affect this? How should the combination of illumination intensity and marking time be optimized?

We have added a section to the Materials and methods titled “Practical considerations for the use of rsCaMPARI” with additional details of illumination wavelengths, intensities, durations as well as other considerations for use. Generally, it is possible to mark the signal over shorter or longer times by using brighter or dimmer photoswitching light, respectively. As shown in Figure 1—figure supplement 9, photoswitching rate is linearly related to light intensity.